# Aminopeptidase Expression in Multiple Myeloma Associates with Disease Progression and Sensitivity to Melflufen

**DOI:** 10.3390/cancers13071527

**Published:** 2021-03-26

**Authors:** Juho J. Miettinen, Romika Kumari, Gunnhildur Asta Traustadottir, Maiju-Emilia Huppunen, Philipp Sergeev, Muntasir M. Majumder, Alexander Schepsky, Thorarinn Gudjonsson, Juha Lievonen, Despina Bazou, Paul Dowling, Peter O`Gorman, Ana Slipicevic, Pekka Anttila, Raija Silvennoinen, Nina N. Nupponen, Fredrik Lehmann, Caroline A. Heckman

**Affiliations:** 1Institute for Molecular Medicine Finland-FIMM, HiLIFE–Helsinki Institute of Life Science, iCAN Digital Precision Cancer Medicine Flagship, University of Helsinki, 00290 Helsinki, Finland; juho.miettinen@helsinki.fi (J.J.M.); romika.kumari@helsinki.fi (R.K.); maiju-emilia.huppunen@helsinki.fi (M.-E.H.); philipp.sergeev@helsinki.fi (P.S.); muntasir.mamun@helsinki.fi (M.M.M.); 2Stem Cell Research Unit, Biomedical Center, University of Iceland, 101 Reykjavik, Iceland; gutra@hi.is (G.A.T.); schepsky@hi.is (A.S.); tgudjons@hi.is (T.G.); 3Department of Hematology, Helsinki University Hospital Comprehensive Cancer Center, 00290 Helsinki, Finland; juha.lievonen@hus.fi (J.L.); pekka.anttila@hus.fi (P.A.); raija.silvennoinen@helsinki.fi (R.S.); 4Department of Hematology, Mater Misericordiae University Hospital, D07 Dublin, Ireland; despina.bazou@ucd.ie (D.B.); pogorman@mirtireland.com (P.O.); 5Department of Biology, Maynooth University, National University of Ireland, W23 F2H6 Maynooth, Co. Kildare, Ireland; paul.dowling@mu.ie; 6Oncopeptides AB, 111 53 Stockholm, Sweden; ana.slipicevic@oncopeptides.com (A.S.); nina.nupponen@oncopeptides.com (N.N.N.); fredrik.lehmann@oncopeptides.com (F.L.)

**Keywords:** multiple myeloma, aminopeptidase, gene expression, melflufen

## Abstract

**Simple Summary:**

The aims of this study were to investigate aminopeptidase expression in multiple myeloma and to identify the aminopeptidases responsible for the activation of the peptide–drug conjugate melflufen in multiple myeloma. We observed a differential expression of aminopeptidases between relapsed/refractory and newly diagnosed multiple myeloma patients. A higher expression of the aminopeptidase genes *XPNPEP1*, *RNPEP*, *DPP3*, and *BLMH* in multiple myeloma plasma cells was associated with shorter patient overall survival. The peptide–drug conjugate melflufen was particularly active towards plasma cells from relapsed/refractory multiple myeloma patients. Melflufen could be hydrolyzed to its active form by the aminopeptidases LAP3, LTA4H, RNPEP, and ANPEP, all of which are expressed in multiple myeloma. These results indicate critical roles for aminopeptidases in disease progression and the activity of melflufen in multiple myeloma.

**Abstract:**

Multiple myeloma (MM) is characterized by extensive immunoglobulin production leading to an excessive load on protein homeostasis in tumor cells. Aminopeptidases contribute to proteolysis by catalyzing the hydrolysis of amino acids from proteins or peptides and function downstream of the ubiquitin–proteasome pathway. Notably, aminopeptidases can be utilized in the delivery of antibody and peptide-conjugated drugs, such as melflufen, currently in clinical trials. We analyzed the expression of 39 aminopeptidase genes in MM samples from 122 patients treated at Finnish cancer centers and 892 patients from the CoMMpass database. Based on ranked abundance, *LAP3*, *ERAP2*, *METAP2*, *TTP2*, and *DPP7* were highly expressed in MM. *ERAP2*, *XPNPEP1*, *DPP3*, *RNPEP*, and *CTSV* were differentially expressed between relapsed/refractory and newly diagnosed MM samples (*p* < 0.05). Sensitivity to melflufen was detected ex vivo in 11/15 MM patient samples, and high sensitivity was observed, especially in relapsed/refractory samples. Survival analysis revealed that high expression of *XPNPEP1*, *RNPEP*, *DPP3*, and *BLMH* (*p* < 0.05) was associated with shorter overall survival. Hydrolysis analysis demonstrated that melflufen is a substrate for aminopeptidases LAP3, LTA4H, RNPEP, and ANPEP. The sensitivity of MM cell lines to melflufen was reduced by aminopeptidase inhibitors. These results indicate critical roles of aminopeptidases in disease progression and the activity of melflufen in MM.

## 1. Introduction

Multiple myeloma (MM) is a highly heterogeneous disease with complex genetic alterations. Advanced molecular profiling of MM has revealed novel therapeutic targets that may aid in drug development. Nevertheless, MM is still incurable for the majority of patients, and new therapeutic approaches are required to improve patient survival. One of the hallmarks of MM is the excessive production of abnormal immunoglobulins, which makes the myeloma cells heavily dependent on the ubiquitin–proteasome system to maintain protein homeostasis and availability of free amino acids. In general, human tumors are highly dependent on free amino acids for their growth, with disruption of protein turnover shown to induce apoptotic cascades in MM as well as in acute myeloid leukemia (AML) [1]. Due to this dependency, inhibition of the ubiquitin–proteasome system is currently the standard of care and a widely accepted strategy in the treatment of both newly diagnosed (NDMM) and relapsed/refractory multiple myeloma (RRMM) patients [2].

Aminopeptidases are an important group of metalloenzymes that catalyze the hydrolysis of terminal amino acid residues from proteins or peptides and operate as the last step downstream of the ubiquitin–proteasome system. Consequently, they are implicated in several cellular functions, including development, differentiation, cell cycle, DNA repair, signal transduction, and programmed cell death [3]. Around 40 genes encoding aminopeptidases have been identified in the human genome, and their activity in the human bone marrow (BM) and blood cells has been described as early as the 1960s [4,5]. Aminopeptidases are involved in many human disorders, including autoimmune diseases and cancer. For example, aminopeptidase N (ANPEP alias CD13) is abnormally expressed in AML, B-cell acute lymphoblastic leukemia, and chronic lymphoblastic leukemia, where it may be used as a diagnostic or prognostic marker [6,7,8]. ANPEP is also expressed in a subset of Waldenström macroglobulinemia, MM, and plasma cell leukemias [9,10]. ANPEP expression in samples from MM patients has been shown to correlate with shorter overall survival [11,12]. Recently, endoplasmic reticulum aminopeptidase 2 (ERAP2) expression has been associated with a better outcome in patients with bladder cancer treated with immunotherapies [13]. Dipeptidyl peptidase 8 (DPP8) has recently been described as a novel target for MM and AML therapeutics [14,15].

Even though aminopeptidases have conserved structures and their role in diseases is well recognized, there are relatively few therapeutic molecules directly targeting these enzymes. Bestatin was the first aminopeptidase inhibitor used in clinical settings, while more recently, a new-generation inhibitor, tosedostat (CHR-2797), entered in clinical trials for AML and myelodysplastic syndromes [16,17]. An alternative approach to directly targeting aminopeptidases is to exploit their mode of action in cells. This is based on observations that aminopeptidase expression may differ between normal and malignant cells or even among different subcellular compartments within a cell. The effects of antibody–drug conjugates or peptide–drug conjugates, such as melphalan flufenamide (hereinafter named melflufen), are facilitated by protease/peptidase hydrolysis, which leads to the release of a highly toxic payload inside the target cells [18,19]. Aminopeptidases have been shown to be one of the peptidase groups facilitating melflufen activation in cells [19]. Melflufen has been shown to have antitumor activity against MM cell lines and primary MM cells [20,21].

To further understand the role of aminopeptidases in MM and disease progression, we investigated the expression of aminopeptidase gene family members in a large cohort of samples from MM patients and assessed differences in expression between NDMM and RRMM. Furthermore, we determined whether specific aminopeptidases are associated with poor prognosis in patients with MM. Using BM cells from MM patients, we observed remarkable ex vivo sensitivity of MM plasma cells to melflufen, particularly in samples from RRMM patients. We were also able to demonstrate that melflufen is a substrate of leucine aminopeptidase 3 (LAP3), leukotriene A4 hydrolase (LTA4H), arginyl aminopeptidase (RNPEP), and ANPEP, which we also found to be expressed in MM.

## 2. Materials and Methods

### 2.1. Sample Collection

Patient samples were collected after informed consent and using protocols approved by an ethical committee of the Helsinki University Hospital (permit numbers 239/13/03/00/2010, 303/13/03/01/2011) in accordance with the Declaration of Helsinki. In total, 178 (NDMM, *n* = 57; RRMM, *n* = 121) different patient samples were collected from 140 different MM patients (Table 1, Online Appendix A). Bone marrow mononuclear cells (BM-MNCs) were isolated from BM aspirates by Ficoll-Paque gradient centrifugation (GE Healthcare, Little Chalfont, Buckinghamshire, UK). For RNA and exome sequencing, BM CD138+ plasma cells were enriched by immunomagnetic bead selection (StemCell Technologies, Vancouver, BC, Canada). 

### 2.2. RNA Sequencing and Analysis

RNA was extracted from CD138+ plasma cells using the AllPrep^®^ DNA/RNA/miRNA Universal or miRNeasy kits (Qiagen, Hilden, Germany). RNA integrity was measured on an Agilent Bioanalyzer 2100 instrument (Agilent, Santa Clara, CA, USA); only samples with RNA integrity ≥7 were used for sequencing. Illumina-compatible RNA sequencing libraries were prepared using ScriptSeq^TM^ technology and sequenced on Illumina HiSeq^®^ 1500 or 2500 instruments (Illumina, San Diego, CA, USA). After preprocessing, filtered reads were aligned to the GRCh38 human reference genome using the STAR aligner tool [22]. Gene read counts were normalized using the reads per kilobase of transcript per million mapped reads (RPKM) method. In total, 39 annotated aminopeptidase genes (Online Appendix A) were identified in the human genome (assembly GRCh38) utilizing the Ensembl release 99 and NCBI databases by using the search term “aminopeptidase” and further confirming the molecular function (gene ontology) of identified genes. A cutoff value >1 RPKM was used to filter the expressed aminopeptidase genes. The DESeq2 tool was used to identify differentially expressed genes in samples from newly diagnosed multiple myeloma (NDMM) vs. relapsed/refractory multiple myeloma (RRMM) [23].

The association of aminopeptidase gene expression with survival outcome was estimated by Kaplan–Meier analysis; the analysis was performed using expression-based grouping of the samples, whereby samples were grouped into “high” (≥median expression) and “low” (<median expression) expression groups. The significance of the difference between the two groups (high vs. low expression) was deduced using a Mantel–Cox logrank test.

### 2.3. Data Validation Using the CoMMpass Dataset

To validate our results, clinical, gene expression, and genomic variant data (somatic mutation and copy number variation (CNVs)) were obtained from the Multiple Myeloma Research Foundation (MMRF) Relating Clinical Outcomes in MM to Personal Assessment of Genetic Profile (CoMMpass) study (both https://research.themmrf.org and www.themmrf.org, were accessed on 5 February 2018).

The MMRF CoMMpass gene expression dataset included 892 samples in total (87% baseline/diagnosis, 12% progressive disease, and 1% missing): 875 bone marrow samples and 17 peripheral blood samples. The MMRF CoMMpass samples were obtained at baseline/diagnosis (*n* = 780), or from patients with progressive disease (*n* = 81), stable disease (*n* = 12), partial response (*n* = 7), and very good partial response (*n* = 4) to treatment; for 8 patients, disease information was missing. A total of 1044 samples (bone marrow, *n* = 1021; peripheral blood, *n* = 23) were used for CNV analysis; samples corresponded to baseline (*n* = 877), progressive disease (*n* = 90), partial response (*n* = 17), stable disease (*n* = 14), very good partial response (*n* = 9), complete response (*n* = 4), and stringent complete response (*n* = 1); information was missing for 32 samples. A total of 1164 samples (bone marrow, *n* = 1140; peripheral blood, *n* = 24) were included in the somatic mutation dataset; samples corresponded to baseline (*n* = 946), progressive disease (*n* = 122), stable disease (*n* = 20), partial response (*n* = 20), very good partial response (*n* = 16), and complete response (*n* = 6); information was missing for 34 samples.

### 2.4. Exome Sequencing and Cytogenetics

The DNeasy^®^ Blood & Tissue kit or AllPrep^®^ DNA/RNA/miRNA Universal kit (Qiagen) was used to isolate genomic DNA from skin biopsies and CD138+ cells. The SeqCap^®^ EZ MedExome kit (Roche NimbleGen, Madison, WI), SureSelect Clinical Research Exome kit, or SureSelect Human All Exon V5 kit (Agilent Technologies, Santa Clara, CA) was used for exome capture. Sequencing was performed on a HiSeq^®^ 1500 or 2500 instrument. VarScan2 somatic algorithm was implemented for calling somatic mutations [24], and mutation annotations were performed using SnpEff 4.04 [25]. Gene CNVs were identified using the CopyCat tool (https://github.com/chrisamiller/copycat, accessed on 19 September 2014). Cytogenetic data were generated using routine diagnostic fluorescence in situ hybridization technology following European Myeloma Network 2012 guidelines [26].

### 2.5. Liquid Chromatography–Tandem Mass Spectrometry-Based Proteomics

BM-MNC CD138+ cells from 23 MM patient samples were lysed in RIPA buffer (Cell Signaling Technology, Danvers, MA, USA), and proteins digested. An amount of 500 ng of each digested whole-cell protein lysate was loaded onto a Q Exactive (Thermo Fisher Scientific, Hemel Hempstead, UK) high-resolution accurate mass spectrometer connected to a Dionex Ultimate 3000 (RSLCnano) chromatography system (Thermo Fisher Scientific, Hemel Hempstead, UK). Peptides were separated using a 2% to 40% gradient of acetonitrile on a Biobasic C18 Picofrit column (Thermo Fisher Scientific, Hemel Hempstead, UK) (100 mm length, 75 mm internal diameter) for over 65 min at a flow rate of 250 nL/min. Data were acquired with the mass spectrometer (MS) operating in automatic data-dependent switching mode. A full MS scan at 140,000 resolution and a range of 300–1700 m/z was followed by an MS/MS scan at 17,500 resolution and a range of 200–2000 m/z, selecting the 10 most intense ions prior to MS/MS.

### 2.6. Proteomics Data Analysis

Protein identification and label-free quantification normalization of MS/MS data were performed using MaxQuant v1.5.2.8 (http://www.maxquant.org, accessed on 20 May 2015). The Andromeda search algorithm incorporated in the MaxQuant software was used to correlate MS/MS data against the *Homo sapiens* UniProt reference proteome database (release 2016_11) and a contaminant sequence set provided by MaxQuant. Perseus v.1.5.6.0 (www.maxquant.org/, accessed on 1 June 2016) was used for data analysis, processing, and visualization. Normalized label-free quantification intensity values were used as the quantitative measurement of protein abundance for subsequent analysis. The data matrix was filtered for the removal of contaminants and peptides identified by site. Label-free quantification intensity values were log2 transformed.

### 2.7. Flow Cytometry-Based Drug Sensitivity Testing

For flow cytometry (FC)-based drug testing, selinexor, 4-hydroperoxycyclophosphamide, and bortezomib (Online Appendix A) were dispensed into 96-well V-bottom plates (Thermo Fisher Scientific, Carlsbad, CA, USA) using an acoustic liquid handling device, Echo 550 (Labcyte, Sunnyvale, CA, USA). Melflufen and melphalan were manually pipetted into the 96-well V-bottom plate wells. Drug plate layout and concentrations are presented in Online Appendix A. Viably frozen BM-MNCs were thawed, freezing solution washed away, cell pellet suspended in conditioned medium (RPMI 1640 medium supplemented with 10% fetal bovine serum (FBS), 2 mM L-glutamine, penicillin (100 U/mL), streptomycin (100 μg/mL), and 25% conditioned medium from the HS-5 human bone marrow stromal cell line), DNase I (Promega, Madison, WI, USA) treated for 60 min, and cultured in conditioned medium overnight. Cells were filtered through a 70 μm cell strainer (Fisher Scientific, Pittsburg, PA), cell viability measured, and cells plated in parallel on pre-drugged 96-well plates (100,000 viable cells/well in 100 μL). If cell viability was poor (<50%), the Dead Cell Removal kit (Miltenyi Biotec, Auburn, CA, USA) was used prior to plating the cells. Stock solutions (10,000 μM) of melflufen (Recipharm AB, Stockholm, Sweden) and melphalan (Sigma-Aldrich, St. Louis, MO, USA) in DMSO, which had been prepared in advance and stored at −80° C, were thawed and diluted in conditioned medium to desired concentrations shortly before the drug testing to minimize autohydrolysis of melflufen and melphalan (exchange of Cl with OH at the *N*-mustard part of the molecules). Melflufen and melphalan were added to the 96-well plates by manually pipetting only after the cells had been plated, just prior to the start of the drug sensitivity test. The cells were incubated with the drugs for 72 h at 37 °C and 5% CO_2_. Following 72 h incubation with the drugs, the cells were centrifuged (500× *g*, 6 min) in the 96-well plates, and media discarded by inverting the plate. The cells were suspended in 25 μL of antibody mix containing staining buffer (5% FBS in Dulbecco’s phosphate-buffered saline) and the following two antibodies (BD Biosciences; Santa Jose, CA, USA): CD138 (BV605, clone MI15, dilution 1:100) and CD38 (BV786, clone HIT2, dilution 1:100). The cells were stained for 30 min at room temperature in the dark and subsequently washed with 100 μL staining buffer followed by centrifugation (500× *g*, 6 min) and supernatant removal by inverting the plate. Apoptotic and dead cells were discriminated by 7-aminoactinomycin D (7-AAD) and PE Annexin V (BD Biosciences) staining with both dyes diluted 1:50 in 25 μL annexin V binding buffer. The plates were incubated for 10 min at room temperature before FC analysis. FC analysis was performed using the IntelliCyt iQue Screener PLUS instrument (Sartorius, Goettingen, Germany). MM patient sample BM-MNCs were acquired for analysis from each 96-well plate well by the iQue Screener PLUS instrument using a 17 s sip time per well and a pump speed of 32 revolutions per minute, resulting in 35 min reading time for a full 96-well plate. ForeCyt software (Sartorius) was used to gate cells and acquire population counts. Analysis was done from viable singlet cells, and the detailed gating strategy is illustrated in Online Appendix A. The cell count of a well was normalized to its six adjacent DMSO controls, from which the highest and lowest values were excluded and normalized values were used for calculating cell viability percentages for each tested drug concentration, and dose-response curves drawn. Half-maximal effective concentration (EC50) values were calculated based on the dose-response curves. 

### 2.8. Cell Lines and Viability Assay

RPMI-8226 (ATCC^®^ CCL-155™) and MM.1S (ATCC^®^ CRL-2974™) cell lines were purchased from the American Type Culture Collection (ATCC; Wesel, Germany). Both cell lines were cultured in RPMI-1640 medium (Gibco, Thermo Fisher Scientific, Waltham, MA, USA), supplemented with 10% heat-inactivated fetal bovine serum (FBS; Gibco, Thermo Fisher Scientific; Waltham, MA, USA), 100 U/mL penicillin, and 100 µg/mL streptomycin (Gibco, Thermo Fisher Scientific, Waltham, MA, USA) at 37 °C in a humidified incubator with 5% CO_2_.

The resazurin-based PrestoBlue cell viability reagent (Thermo Fisher Scientific, Waltham, MA, USA) was used to assess the viability of RPMI-8226 and MM.1S. Cells were treated for 48 h with complete growth medium supplemented with 3.75, 7.5, 15, 30, or 60 μM tosedostat. Then, RPMI-8226 and MM.1S cells were pretreated with DMSO (control) or the metalloaminopeptidase inhibitors bestatin (10 μM) and tosedostat (10 μM) for 1 h, and then either left without additional treatment or treated with three different concentrations (1, 3, and 5 μM) of melflufen/melphalan for 15 min, after which the medium was replaced. Cells were cultured in 96-well culture plates at a density of 20,000 cells/well (250 cells/μL), incubated for 48 h at 37 °C in a humidified incubator with 5% CO_2_, after PrestoBlue was added (1/10 th of the total volume). After 2 h incubation at 37 °C in a humidified incubator with 5% CO_2_, absorbance was read at 570 nm with a reference wavelength of 595 nm. Each experimental setup was performed three times independently with at least three replicates, and values were normalized to untreated control (mean ± standard deviation (SD), *n* = 3). Statistical significance was assessed with one- or two-way analysis of variance (ANOVA) followed by Dunnett’s or Tukey’s multiple comparison tests using GraphPad Prism (* *p* ≤ 0.05, ** *p* ≤ 0.01, *** *p* ≤ 0.001, **** *p* ≤ 0.0001).

### 2.9. Hydrolysis Assay

Frozen solutions containing the aminopeptidases LAP3 (TP309052) (OriGene, Rockville, MD, USA), DPP3 (8087), DPP7 (3438-SE), LTA4H (4008-ZN), RNPEP (8089-ZN), XNPEP1 (2970-ZN), and ANPEP (3815-ZN) (R&D Systems, Minneapolis, MN, USA) were thawed and diluted with their incubation buffers (Online Appendix A) to the desired concentrations. A stock solution (1000 μM) of the test substrate melflufen was prepared in DMSO and further diluted in incubation buffer to the desired concentration shortly before the experiment to minimize autohydrolysis of melflufen. Incubation buffer and substrates were mixed and incubated with aminopeptidases for 2 h at a substrate-to-enzyme ratio of 5000:1. Control incubations were performed without aminopeptidases. All incubations were performed in duplicate at 37 °C. The incubation was stopped by adding ice-cold acetonitrile. The samples were centrifuged and diluted with eluent A (water + formic acid, 1000 + 1, v + v). The samples were subjected to liquid chromatography–high-resolution mass spectrometry analysis. The analysis system consisted of a binary high-performance liquid chromatography (HPLC) pump (1290 Series, Agilent), a column oven at 40 °C (1260 series, Agilent), an autosampler (HTC PAL, CTC Analytics, Zwingen, Switzerland), and an Orbitrap mass spectrometer with electrospray ionization (Q Exactive, Thermo Scientific). A Phenomenex Luna 3μ C18(2) HPLC column (50 × 2 mm, 3 μm) was used with a C18 4 × 2 mm guard column (Phenomenex, Torrance, CA), and a binary gradient of eluents A and B (acetonitrile + formic acid, 1000 + 1, v + v) was applied for separation. Mass spectra were recorded in positive ionization mode at a resolution of 35,000 in FullScan mode. The MS data were evaluated for (M + M)^+^ traces of melflufen, monohydroxy melphalan flufenamide, dihydroxy melphalan flufenamide, and 4‑F‑PheOEt, with peaks integrated using LCquan (Thermo Scientific).

## 3. Results

### 3.1. Differential Expression of Aminopeptidase Genes in MM

To determine relative mRNA expression patterns of aminopeptidases in MM cells, we performed RNAseq analysis on 122 MM patient-derived samples (NDMM = 41, RRMM = 81) from 99 different patients (Online Appendix A). First, we ranked aminopeptidase gene expression levels based on mRNA abundance in all samples and observed that most aminopeptidases were expressed (Figure 1A, Table 2, Online Appendix A). The most abundantly expressed aminopeptidase genes were *LAP3*, *ERAP2*, methionyl aminopeptidase 2 (*METAP2*), tripeptidyl peptidase 2 (*TTP2*), *DPP7*, *ERAP1*, *LTA4H*, and leucyl and cystinyl aminopeptidase (*LNPEP*) with a median log2(RPKM) range of 5.00 (*LAP3*) to 2.93 (*LNPEP*). The least abundantly expressed aminopeptidase genes were thyrotropin-releasing hormone-degrading enzyme (*TRHDE*), F coagulation factor XI (*F11*), X-prolyl aminopeptidase 2 (*XPNPEP2*), cathepsin V (*CTSV*), laeverin (*LVRN*), and archaelysin family metallopeptidase 1 (*AMZ1*) with a median log2(RPKM) range of −6.57 (*TRDHE*) to −3.60 (*AMZ1*). We further verified that aminopeptidase gene expression profiles are comparable between the Multiple Myeloma Research Foundation (MMRF) CoMMpass dataset (*n* = 892) and our internal (FIMM) dataset with a correlation coefficient of 0.93 (*p* = 2.2 × 10^−16^) (Online Appendix A). In the FIMM dataset, aminopeptidases clustered hierarchically into four subgroups, with group I displaying the highest level of gene expression and group IV the lowest (Figure 1B). We observed correlation between expression levels of some aminopeptidase genes in group I, for example, *TPP2* and *DPP7*, and certain cytogenetic markers and with patient age (Online Appendix A). *TPP2* was expressed at a lower level in patients with del 13q and in patients ≥65 years of age. *DPP7* was expressed at a lower level in patients with either t(4;14) or t(11;14), but was more highly expressed in patients ≥65 years of age. 

Liquid chromatography–tandem mass spectrometry-based proteomics data were collected from 23 MM patients (*n* = 14 NDMM; *n* = 9 RRMM) (Online Appendix A). The proteomic results showed a positive correlation with mRNA expression in CD138+ cells enriched from MM patient BM-MNCs. Out of the 39 aminopeptidases, peptides positively identifying 17 of these proteolytic enzymes were detected by mass spectrometry (Online Appendix A, Appendix A). Of these 17 aminopeptidases, six, namely, LAP3, bleomycin hydrolase (BLMH), DPP3, DPP7, RNPEP, and ERAP2, showed the highest correlation between protein and gene expression levels (Online Appendix A).

### 3.2. Association between Aminopeptidase Gene Expression and MM Disease Status

Next, we investigated whether the expression of aminopeptidase genes could be linked to disease status. We compared aminopeptidase gene expression profiles between the 41 NDMM and 81 RRMM patient samples in the FIMM dataset. In total, five aminopeptidases, *ERAP2*, *XPNPEP1*, *DPP3*, *RNPEP*, and *CTSV*, were differentially expressed in the RRMM and NDMM samples (*p* < 0.05; adjusted *p* < 0.1), indicating potential roles in disease progression (Figure 2A). Of the five differentially expressed aminopeptidase genes, only *ERAP2* was decreased in RRMM vs. NDMM samples (Figure 2B). In the MMRF CoMMpass dataset using paired samples (NDMM, *n* = 39; RRMM, *n* = 45), similar differences were observed, although only *CTSV* had an adjusted *p*-value < 0.1 (Online Appendix A). We observed correlation between the expression levels of some of the five differentially expressed aminopeptidase genes in RRMM vs. NDMM, for example, *DPP3*, *RNPEP*, and *XPNPEP1*, and certain cytogenetic markers and with International Staging System (ISS) stage (Online Appendix A). *DPP3* was expressed at a lower level in patients with t(11;14), and also in ISS stage 2 compared with ISS stage 3. *RNPEP* was more highly expressed in patients with either del 13q or t(4;14) or 1q gain, and in ISS stage 3 compared with both ISS stages 1 and 2. *XPNPEP1* was more highly expressed in patients with 1q gain, and in ISS stage 3 compared with ISS stage 2.

### 3.3. Prognostic Significance of Aminopeptidase Gene Expression

High expression of four aminopeptidase genes associated with significantly poorer prognosis in MM patients from both FIMM (Figure 3A–E and Online Appendix A–F) and CoMMpass datasets (Online Appendix A–M). The median overall survival associated with patient samples showing high versus low gene expression was 55 vs. 122 months for *XPNPEP1* (hazard ratio (HR), 3.263; 95% confidence limit (CL), 1.749–6.087; *p* = 0.00012) (Figure 3B), 73 vs. 100 months for *RNPEP* (HR, 2.271; 95% CL, 1.248–4.129; *p* = 0.0062) (Figure 3C), 73 vs. 120 months for *DPP3* (HR, 1.956; 95% CL, 1.108–3.453; *p* = 0.02) (Figure 3D), and 68 vs. 122 months for *BLMH* (HR, 1.815; 95% CL, 1.035–3.184; *p* = 0.037) (Figure 3E, Online Appendix A).

### 3.4. Aminopeptidase Somatic Mutation and CNV Characteristics in Myeloma

Exome sequence analysis of 169 samples (*n* = 56 NDMM and *n* = 113 RRMM) from 132 different MM patients in the FIMM dataset (Online Appendix A) indicated that small somatic variants within the protein coding regions of aminopeptidase genes are relatively rare. In this dataset, *NPEPPS* was mutated in 2.37% of the samples (4/169), *LVRN* in 2.37% of the samples (4/169), and *BLMH* in 1.78% of the samples (3/169), while the frequency of mutations in other aminopeptidases was below 1.2% (Online Appendix A, Appendix A). Mutation frequencies were similarly infrequent in the CoMMpass dataset samples (*n* = 1164), with aminopeptidase mutations identified in less than 1.2% of the samples (Online Appendix A).

CNV analysis showed that gains in the copy number of the aminopeptidase genes *DPP9* (31.95%), *PGPEP1* (27.22%), *RNPEP* (20.12%), *PEPD* (20.12%), and *DPP7* (17.75%) were frequent, whereas deletions of *TPP2* (40.24%), *XPNPEP2* (13.61%), and *XPNPEP3* (8.87%) were observed in more than 8% of the samples (Online Appendix A, Appendix A, Appendix A). Results in the FIMM dataset were validated using the CoMMpass dataset: the correlation coefficient for aminopeptidase gain percentage was 0.73, and for aminopeptidase deletion, the percentage was 0.7 (Online Appendix A). As expected, RRMM samples exhibited a higher number of CNVs associated with aminopeptidase genes than NDMM samples (RRMM: *n* = 467 overall, with gains in 317 and deletions in 150 vs. NDMM: *n* = 159, with gains in 95 and deletions in 64).

### 3.5. MM Patient Bone Marrow CD138+CD38+ Plasma Cells Are Sensitive to Melflufen

Aminopeptidases not only have a role in disease progression, but these enzymes are required to direct the activity of peptide–drug conjugates, such as melflufen. To determine whether melflufen activity can be detected and differs in BM samples from MM patients, we evaluated the efficacy of melflufen and four additional drugs (melphalan, selinexor, bortezomib, and 4-hydroperoxycyclophosphamide) towards CD138+CD38+ plasma cells in 15 BM-MNC samples (NDMM = 6; RRMM = 9) obtained from 14 different MM patients (Online Appendix A, Appendix A). CD138+CD38+ plasma cell-specific responses to the drugs were measured by multiplexed high-throughput FC. The fraction of CD138+CD38+ cells from all live cells in the tested samples varied between 0.4% and 27.9% after 72 h incubation in the ex vivo culturing conditions without drug treatment (Online Appendix A). Of the 5 drugs tested, melflufen showed the highest activity against CD138+CD38+ cells (median EC50 = 0.9 nM) (Online Appendix A). Of the 15 samples tested, 5 were highly sensitive to melflufen (median EC50 <0.1 nM), 6 were intermediately sensitive (median EC50 0.1–5 nM), and the 4 least-sensitive samples had a median EC50 >5 nM (Figure 4A). In general, CD138+CD38+ cells were more sensitive to melflufen than melphalan (median EC50 = 1473 nM) (Online Appendix A). There was a positive correlation between melflufen and melphalan response in CD138+CD38+ cells (r = 0.66; *p* = 0.009) (Figure 4B). We did not observe correlation between myeloma patient sample CD138+CD38+ cell melflufen sensitivity and aminopeptidase gene expression in a small set of 10 myeloma samples (Online Appendix A).

Of the other tested drugs, selinexor, a XPO1/CRM1 inhibitor, also showed high toxicity against CD138+CD38+ cells (median EC50 = 35.4 nM) (Online Appendix A, Appendix A). MM sample sensitivity to selinexor did not correlate with melflufen sensitivity or any of the other tested drugs (Online Appendix A–C). Bortezomib, a proteasome subunit S26 inhibitor, showed a very narrow window of response in CD138+CD38+ cells in the ex vivo assay with a median EC50 = 5.7 nM (Online Appendix A, Appendix A). There were five samples that could be considered sensitive to bortezomib (median EC50 = 4.2 nM) and three samples insensitive (median EC50 = 12.3 nM). None of the samples showed sensitivity to 4-hydroperoxycyclophosphamide, the active metabolite of cyclophosphamide, with an EC50 ≥1900 nM in the ex vivo assay (Online Appendix A, Appendix A).

### 3.6. RRMM Samples Are More Sensitive to Melflufen than NDMM Samples

Interestingly, our FC-based drug sensitivity testing results suggested that CD138+CD38+ cells from RRMM patients were more sensitive ex vivo to melflufen compared with CD138+CD38+ cells in NDMM samples (RRMM median EC50 = 0.04 nM; NDMM median EC50 = 7.7 nM; *p* = 0.0004) (Figure 4C). A similar trend was observed with melphalan, although the median EC50 values were clearly higher (RRMM median EC50 = 556 nM; NDMM median EC50 = 3193 nM; *p* = 0.025) when compared with melflufen (Figure 4C). CD138+CD38+ cells from patients with poor prognosis indicators, such as cytogenetic markers del 17p and t(4;14), included both melflufen-sensitive (*n* = 5) and insensitive (*n* = 5) samples (Online Appendix A). No clear biomarkers for sensitivity to melflufen were identified based on cytogenetic markers alone. Both melflufen-sensitive and insensitive samples expressed aminopeptidase genes (Figure 1B).

### 3.7. Melflufen Activity Is Dependent on Aminopeptidase Activity 

To determine whether melflufen activity is dependent on aminopeptidase activity, MM cell lines RPMI-8226 and MM.1S were first treated with the metalloaminopeptidase inhibitors bestatin and tosedostat, then treated with melflufen or melphalan, and the viability of cells measured after 48 h. The viability of RPMI-8226 and MM.1S cells decreased in a dose-dependent manner upon 48 h treatment with tosedostat alone (Figure 5A). Treating RPMI-8226 cells with 10 μM tosedostat prior to treatment with melflufen significantly increased the viability of the cells from 60.41% to 78.15% with 3 μM melflufen (*p* = 0.0086) and from 21.77% to 54.55% with 5 μM melflufen (*p* ≤ 0.0001), compared with untreated controls (Figure 5B). Likewise, treating MM.1S cells with 10 μM tosedostat prior to treatment with melflufen significantly increased the viability of the cells from 39.82% to 73.78% for 0.5 μM melflufen (*p* = 0.0175) and from 19.81% to 51.60% for 1 μM melflufen (*p* = 0.0046) compared with untreated control cells (Figure 5C). In addition, treating RPMI-8226 and MM.1S cells with bestatin reduced their sensitivity to melflufen, albeit not significantly. 

### 3.8. Aminopeptidases LAP3, LTA4H, RNPEP, DPP7, and ANPEP Hydrolyze Melflufen

Melflufen can be hydrolyzed to melphalan and para-fluoro-L-phenylalanine ethyl ester (4-F-Phe-OEt) under suitable conditions (Figure 6A, Online Appendix A, Online Appendix A). We tested seven aminopeptidases, DPP3, DPP7, XPNPEP1, LAP3, LTA4H, RNPEP, and ANPEP, to determine whether they induce melflufen amide hydrolysis in vitro. The release of 4-F-Phe-OEt from melflufen was measured after incubating melflufen with the aminopeptidases for 2 h. Release of 4-F-Phe-OEt was measured using liquid chromatography–high-resolution mass spectrometry, which showed that the peak area of melflufen decreases while the peak area for 4-F-Phe-OEt increases when a specific aminopeptidase is added. A clear increase of 4-F-Phe-OEt was observed after melflufen incubation with aminopeptidases LAP3 (Figure 6B), LTA4H (Figure 6C), RNPEP (Figure 6D), and ANPEP (Online Appendix A), suggesting that these aminopeptidases can hydrolyze melflufen. However, no such increase was observed with DPP3 and XPNPEP1. Interestingly, aminopeptidase DPP7 induced hydrolysis of the terminal ester and not the amide of the melflufen molecule. The DPP family of enzymes hydrolyzes the peptide bond between two amino acids from the N-terminal. Due to the structure of melflufen, it is likely that DPP7 hydrolyzes the ester rather than the amide bond.

## 4. Discussion

Aminopeptidases are ubiquitous enzymes with important roles in cell development, growth, and maintenance. They are widely expressed in tissues, including hematopoietic cells of the bone marrow and peripheral blood, and in lymphoid organs [27,28]. Aminopeptidases are thought to have basic roles in regulating cell homeostasis, but more specific functions have recently been identified, particularly in malignancies [1,29]. The utilization of aminopeptidases for the processing of novel therapeutic drugs, such as antibody–drug conjugates and peptide­–drug conjugates, is an active research area in drug development, especially for plasma cell diseases. The chemical linker in melflufen is made of a dipeptide, which is a substrate for aminopeptidases, resulting in hydrolysis and rapid release of the cytotoxic alkylator payload and intermediate metabolites [30]. Therefore, the expression and activity of aminopeptidases in MM cells is of particular interest.

Analysis of gene expression profiles of CD138+ cells from MM patient samples from two datasets showed that the majority of aminopeptidases are expressed at relatively similar levels. LAP3, one of the most highly expressed aminopeptidases in MM, is an enzyme that primarily catalyzes the hydrolysis of leucine residues from the amino-termini of proteins or peptides [31]. LAP3 is also implicated in antigen and major histocompatibility complex class I peptide processing, and is associated with the proinflammatory effects of IFN-γ [32]. In cancer, LAP3 has also been linked to tumor cell proliferation, migration, and invasion [33,34]. Altered *LAP3* expression has also been observed in esophageal squamous cell and hepatocellular carcinoma [35,36]. The role of LAP3 in plasma cells and other hematological cells is currently not known. 

Only five aminopeptidases, namely, *ERAP2*, *XPNPEP1*, *DPP3, RNPEP*, and *CTSV,* were differentially expressed between NDMM and RRMM. In the endoplasmic reticulum, ERAP enzymes trim peptides for presentation on major histocompatibility complex class I molecules [37]. A small number of studies demonstrate the direct involvement of ERAP enzymes in tumor growth and the generation of tumor epitopes for the presentation of cytotoxic T cells [38]. For example, the absence of *ERAP2* in choriocarcinoma has been linked to the reduction of the activation of T lymphocytes by the tumor cells [39]. One could speculate that the reduction of *ERAP2* expression in myeloma cells could be beneficial by reducing their immunogenicity and leading to their increased survival by immune evasion. The selection for lower *ERAP2*-expressing cells would naturally occur during disease progression. Another hypothetical link between MM and *ERAP2* could be the excessive plasma protein production in MM cells that induces endoplasmic reticulum stress. The proteasomal degradation of the paraprotein is required to avoid endoplasmic reticulum stress-induced cell death. If this machinery is inhibited or altered, it might lead to drug resistance. In addition, *CTSV* is a potential prognostic biomarker for progression with high *CTSV* expression shown to be associated with poor outcome in breast ductal carcinomas in situ [40].

We also found that elevated expression of the aminopeptidase genes *XPNPEP1*, *RNPEP*, *DPP3*, and *BLMH* could predict poor survival of MM patients. The role of these and other highly expressed aminopeptidases in MM remains to be explored. Proline-specific DPPs are emerging drug targets, and DPP4 inhibitors (gliptins) are already approved globally for the treatment of diabetes. Inhibition of DPP7 leads to apoptosis of the majority of resting cells in chronic lymphocytic leukemia [41]. Another study found that inhibition of DPP7 leads to upregulation of *RB1* and *MYC* and decrease of *BCL2*, leading to potential apoptosis induction [42]. However, *TP53* inactivation prevents *DPP7* downregulation [42]. Biallelic inactivation of *TP53* is a hallmark of MM and could therefore explain *DPP7* overexpression in MM [43,44]. *DPP3* is overexpressed in estrogen receptor-positive breast cancer and correlates with poor prognosis, and it is also more highly expressed in endometrial carcinomas vs. normal tissue [45,46]. Off-target DPP8/9 inhibition in MM cells leads to apoptotic cell death signaling in the presence of DPP4 inhibitors (gliptins), whereas in AML the direct inhibition of DPP8/9 induces the pro-caspase-1-dependent inflammatory form of programmed cell death, pyroptosis [14,15].

There are only a very limited number of studies demonstrating the prognostic value of aminopeptidases, namely, BLMH, RNPEP, and XPNPEP1, in human tumors. BLMH may have multiple roles in different physiological and pathological conditions, such as resistance to bleomycin therapy in several cancer types, cellular detoxification, and training of peptides for antigen presentation [47,48]. The XPNPEP1 (alias aminopeptidase P) protein is specifically found in the blood vessels of lungs and also in lung tumors [49]. *RNPEP* (alias aminopeptidase B) mRNA levels have been demonstrated to be altered during colorectal adenoma–carcinoma evolution, and RNPEP plasma levels are also independent prognostic factors for colorectal cancer patients [50]. Both XPNPEP1 and RNPEP enzyme activities were found to be significantly elevated in thyroid neoplasms when compared with nonmalignant adjacent tissues [51].

Peptide–drug conjugates such as melflufen require aminopeptidases to unload their payload into cells. We found that melflufen could target CD138+CD38+ plasma cells in MM patient samples and was particularly active in RRMM samples. The higher expression of specific aminopeptidases in RRMM compared with NDMM could explain the differential activity of melflufen in samples from these two patient groups. However, from our limited set of samples, we did not observe significant correlation between melflufen sensitivity and expression of an individual aminopeptidase gene. Nevertheless, hydrolysis analysis of seven aminopeptidases demonstrated that LAP3, LTA4H, RNPEP, DPP7, and ANPEP are able to hydrolyze melflufen. In addition, we demonstrated that the efficacy of melflufen, but not melphalan, could be reduced with the addition of the metalloaminopeptidase inhibitors tosedostat or bestatin in MM cells lines. This indicates a difference in the mode of action between melphalan and melflufen. 

Melflufen is being investigated in several clinical trials. The first study reported shows that melflufen combined with dexamethasone leads to clinical improvement with an overall response rate of 31%, median progression-free survival of 5.7 months, and median overall survival of 20.7 months [52]. It also demonstrated that melflufen is equally effective in melphalan-naïve and melphalan-refractory patients [52]. The second study, focusing on heavily pretreated RRMM, showed that melflufen combined with dexamethasone leads to clinical improvement in the triple-class RRMM population with an overall response rate of 26%, median progression-free survival of 3.9 months, and median overall survival of 11.2 months [53]. In this study, we demonstrated that melflufen is highly effective at targeting plasma cells, particularly in samples from RRMM patients, including those with poor prognostic markers, such as del 17p or t(4;14). We also showed that melflufen can be hydrolyzed to its active form by aminopeptidases that are highly expressed in MM. Although additional investigations are needed to understand the specific role of aminopeptidases in MM pathogenesis, our results highlight that this group of enzymes can be exploited to benefit MM patients who may not respond to other available therapies.

## 5. Conclusions

In this study we have analyzed the expression of aminopeptidase genes in MM patients and how aminopeptidase gene expression correlates with patient overall survival. We also tested the sensitivity of MM plasma cells to peptide–drug conjugate melflufen, which is known to be activated by aminopeptidases, and how inhibition of aminopeptidases affect melflufen activity in MM cell lines. Based on ranked abundance, *LAP3*, *ERAP2*, *METAP2*, *TTP2*, and *DPP7* were highly expressed in MM. *ERAP2*, *XPNPEP1*, *DPP3*, *RNPEP*, and *CTSV* were differentially expressed between RRMM and NDMM samples (*p* < 0.05). Survival analysis revealed that high expression of *XPNPEP1*, *RNPEP*, *DPP3*, and *BLMH* (*p* < 0.05) was associated with shorter overall survival. Sensitivity to melflufen was detected ex vivo in the majority of MM patient samples, and high sensitivity was observed, especially in RRMM samples. Hydrolysis analysis demonstrated that melflufen is a substrate for aminopeptidases LAP3, LTA4H, RNPEP, and ANPEP, which are expressed in MM. The sensitivity of MM cell lines to melflufen was reduced by aminopeptidase inhibitors. These results indicate critical roles of aminopeptidases in disease progression and the activity of melflufen in MM.

## Figures and Tables

**Figure 1 cancers-13-01527-f001:**
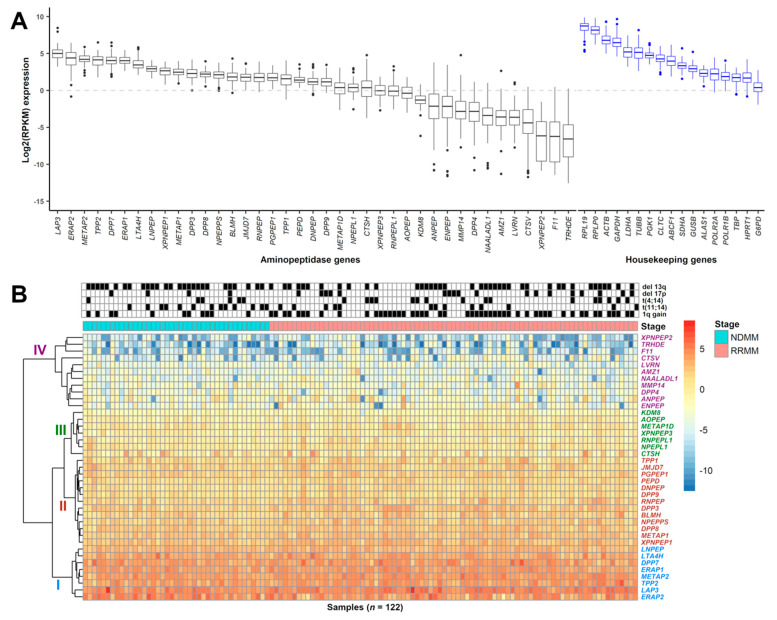
Expression of aminopeptidase gene family in MM patient CD138+ cells. (**A**) Boxplot showing the log2(RPKM) expression of 39 aminopeptidase and 17 housekeeping genes from 122 MM samples (NDMM (*n* = 41), RRMM (*n* = 81)) taken from 99 different patients. Genes are ranked based on median expression values. (**B**) Heatmap showing the hierarchical clustering of aminopeptidase gene expression, with group I (blue) containing genes with the highest and group IV (magenta) containing genes with the lowest level of expression. Groups II (red) and III (green) contain genes with an intermediate level of expression. Sample cytogenetics (del 13q, del 17p, t(4;14), t(11;14), 1q gain) are indicated above the heatmap. Disease stage indicates whether the sample was from NDMM (shown as cyan; samples on the left) or RRMM (pink; samples on the right). RPKM: reads per kilobase of transcript per million mapped reads; MM: multiple myeloma; NDMM: newly diagnosed multiple myeloma; RRMM: relapsed/refractory multiple myeloma.

**Figure 2 cancers-13-01527-f002:**
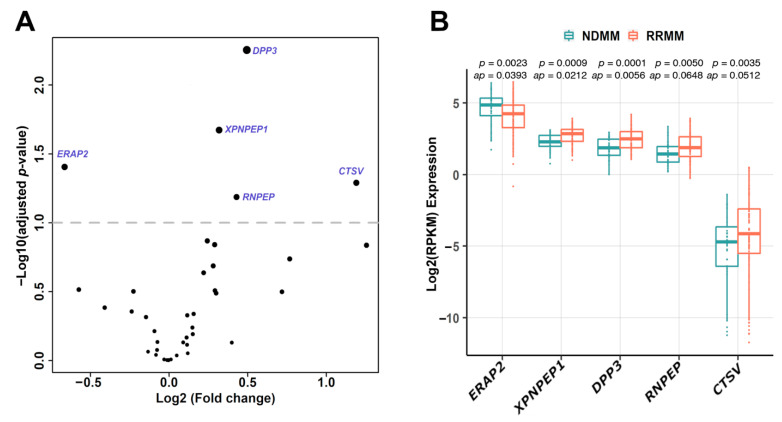
Aminopeptidase genes are differentially expressed in NDMM versus RRMM. (**A**) Five aminopeptidase genes (*ERAP2*, *XPNPEP1*, *DPP3*, *RNPEP*, *CTSV*) were differentially expressed in RRMM (*n* = 82) and NDMM (*n* = 41) samples (*p* < 0.05; adjusted *p* < 0.1) in the FIMM dataset. Differential gene expression between NDMM and RRMM was determined using the DESeq2 tool. (**B**) Among these five genes, only the expression level of *ERAP2* was decreased in RRMM versus NDMM samples. NDMM: newly diagnosed multiple myeloma; RRMM: relapsed/refractory multiple myeloma; RPKM: reads per kilobase of transcript per million mapped reads; *ap*: adjusted *p*-value.

**Figure 3 cancers-13-01527-f003:**
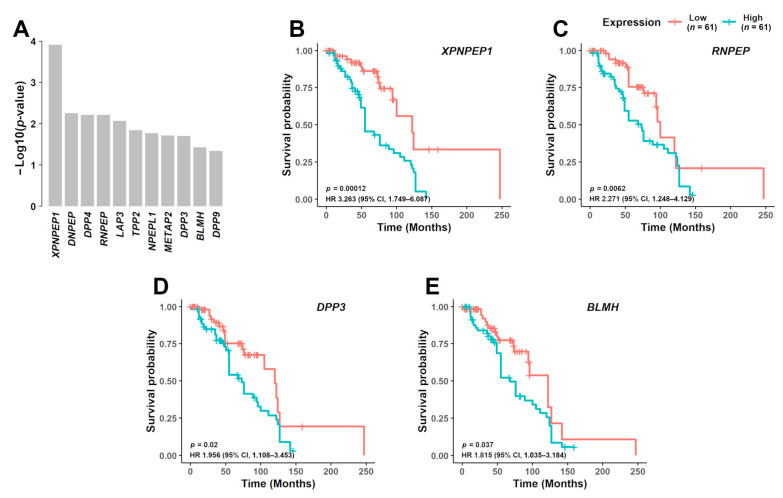
Prognostic significance of aminopeptidase gene expression in MM. (**A**) In the FIMM dataset consisting of 122 MM samples, NDMM (*n* = 41) and RRMM (*n* = 81), from 99 different patients, 11 aminopeptidase genes were predicted to be associated with poor prognosis (*p* < 0.05). From these 11 genes, 4 were further validated in the CoMMpass dataset. Survival curves are shown for the 4 aminopeptidase genes based on the FIMM dataset, dividing the patients with low gene expression (*n* = 61) and high gene expression (*n* = 61): (**B**) *XPNPEP1*, (**C**) *RNPEP*, (**D**) *DPP3*, and (**E**) *BLMH*. NDMM: newly diagnosed multiple myeloma; RRMM: relapsed/refractory multiple myeloma; HR: hazard ratio; CI: confidence interval.

**Figure 4 cancers-13-01527-f004:**
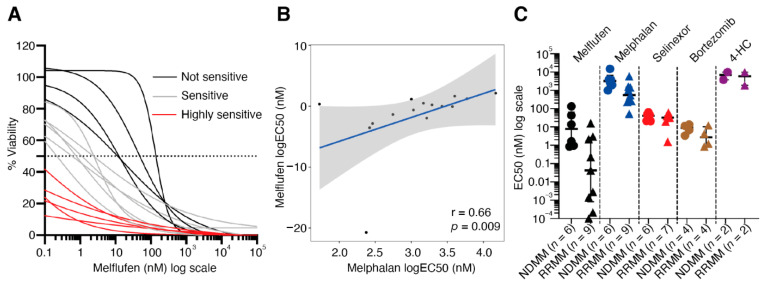
MM patient bone marrow CD138+CD38+ plasma cells are sensitive to melflufen. (**A**) Drug sensitivity was measured after 72 h incubation with melflufen or with DMSO alone (control). Cell viability was assessed by multicolor high-throughput flow cytometry using annexin V and 7AAD viability markers. The viability (%) of CD138+CD38+ cells was calculated, and dose-response curves drawn. The MM patient samples can be divided into three groups based on the sensitivity of their CD138+CD38+ cells to melflufen: highly sensitive (red: EC50 < 0.1 nM), intermediately sensitive (gray: EC50 0.1–5 nM), not sensitive (black: EC50 > 5 nM). (**B**) Correlation of CD138+CD38+ cell sensitivity to melflufen (*y*-axis) with melphalan sensitivity (*x*-axis). Drug sensitivity was measured using EC50 values. (**C**) Comparison of CD138+CD38+ cell sensitivity (EC50) between NDMM (dots) and RRMM (triangles) in patient samples treated with melflufen (black; *p* = 0.0004), melphalan (blue; *p* = 0.025), selinexor (red; *p* = 0.44), bortezomib (brown; *p* = 0.34), and 4-HC (purple; *p* = 1). DMSO: dimethyl sulfoxide; 4-HC: 4-hydroperoxycyclophosphamide; NDMM: newly diagnosed multiple myeloma; RRMM: relapsed/refractory multiple myeloma.

**Figure 5 cancers-13-01527-f005:**
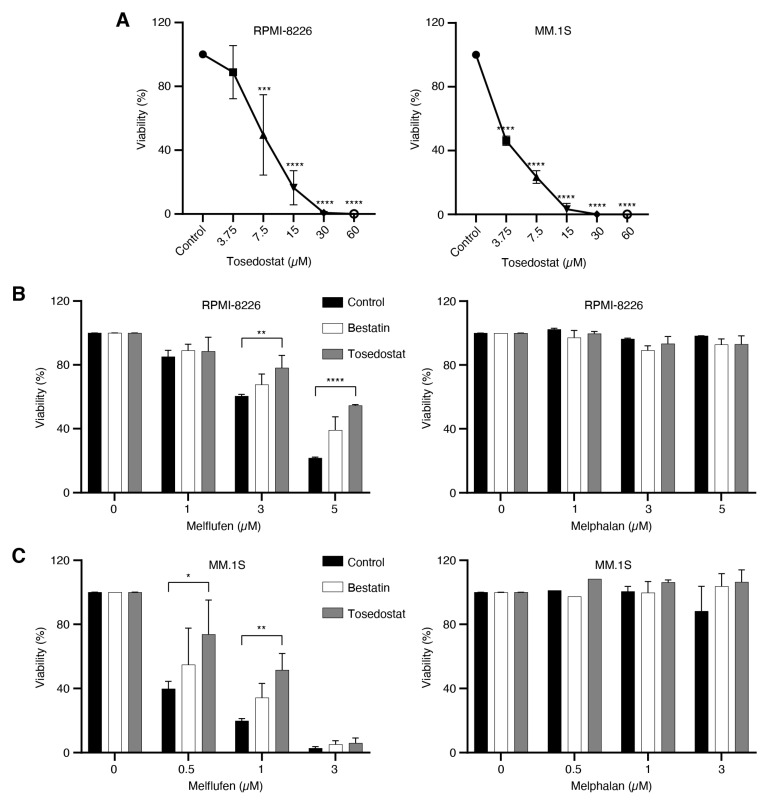
Aminopeptidase inhibitors reduce sensitivity to melflufen in MM cell lines. (**A**) MM cell lines RPMI-8226 and MM.1S were treated with different concentrations of the aminopeptidase inhibitor tosedostat for 48 h, and cell viability was measured using PrestoBlue cell viability reagent. (**B**) RPMI-8226 cells or (**C**) MM.1S cells were pretreated with DMSO (control) or the aminopeptidase inhibitors bestatin (10 μM) or tosedostat (10 μM) for 1 h, then either left without additional treatment or treated with three different concentrations of melflufen or melphalan for 15 min, after which the medium was replaced. After 48 h, cell viability was measured using PrestoBlue cell viability reagent. Statistical significance is indicated as **p* ≤ 0.05; ***p* ≤ 0.01; ****p* ≤ 0.001; *****p* ≤ 0.0001. DMSO: dimethyl sulfoxide

**Figure 6 cancers-13-01527-f006:**
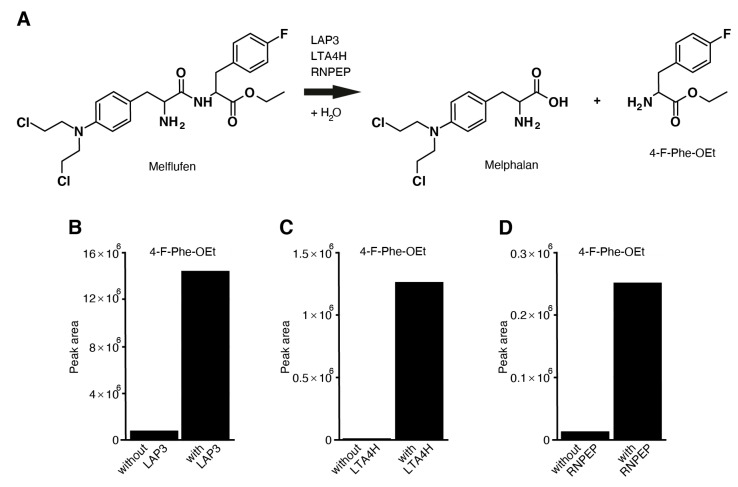
Aminopeptidases LAP3, LTA4H, and RNPEP can hydrolyze melflufen to melphalan and 4-F-Phe-OEt. (**A**) Reaction pathway of the hydrolysis of melflufen peptide bond by the aminopeptidases LAP3, LTA4H, and RNPEP. (**B**–**D**) Incubation buffer and substrates were mixed and incubated without and with (**B**) LAP3, (**C**) LTA4H, and (**D**) RNPEP for 2 h at 37 °C (with duplicate incubations). The incubation was stopped by adding ice-cold acetonitrile. Using LC-HRMS, 4-F-Phe-OEt was then quantified, with concentrations presented as the peak area in the figure. 4-F-Phe-OEt: para-fluoro-L-phenylalanine ethyl ester; LC-HRMS: liquid chromatography–high-resolution mass spectrometry.

**Table 1 cancers-13-01527-t001:** MM patient and Disease Characteristics and Treatment History in the FIMM Dataset.

Patient and disease characteristics by disease stage ^a^			
	NDMM (*n* = 57)	RRMM (*n* = 83)	Total (*n* = 140)
Age at diagnosis, years, median (range)	65 (46–84)	63 (26–81)	64 (26–84)
Sex, female/male, *n*	27/30	31/52	58/82
Cytogenetics, *n* (%)			
t(11;14)	15 (26.3)	16 (19.3)	31 (22.1)
t(4;14)	9 (15.8)	19 (22.9)	28 (20.0)
t(14;16)	2 (3.5)	2 (2.4)	4 (2.9)
t(14;20)	0	2 (2.4)	2 (1.4)
del(17p)	5 (8.8)	20 (24.1)	25 (17.9)
del(13q)/-13	39 (68.4)	42 (50.6)	81 (57.9)
1q gain	18 (31.6)	46 (55.4)	64 (45.7)
No abnormalities found	2 (3.5)	0	2 (1.4)
International Staging System, *n* (%)			
1	11 (19.3)	19 (22.9)	30 (21.4)
2	27 (47.4)	23 (27.7)	50 (35.7)
3	11 (19.3)	16 (19.3)	27 (19.3)
Not available	8 (14.0)	25 (30.1)	33 (23.6)
**Treatment history of relapsed/refractory patients (*n* = 83)**			
Prior treatment, *n* (%)	Exposed	Refractory	Not exposed
Alkylating agents (MEL, CPM)	63 (75.9)	16 (19.3)	4 (4.8)
Bortezomib	44 (53.0)	28 (33.7)	11 (13.3)
IMiDs	31 (37.3)	34 (41.0)	18 (21.7)

^a^ If a patient provided both NDMM and RRMM samples, this patient was included in the NDMM group. If a patient provided samples at multiple relapse stages and the diagnosis sample was missing, then data from the first relapse were included in the table. FIMM: Institute for Molecular Medicine Finland; NDMM: newly diagnosed multiple myeloma; RRMM: relapsed/refractory multiple myeloma; MEL: melphalan; CPM: cyclophosphamide; IMiDs: immunomodulatory drugs.

**Table 2 cancers-13-01527-t002:** List of evaluated aminopeptidase genes (*n* = 39), sorted by median log2(RPKM) expression from RNAseq data from 122 multiple myeloma patient samples in the FIMM dataset.

Gene Symbol ^a^	Genomic Location	Median log2(RPKM) Expression (*n* = 122)	Gene Name ^a^	Peptidase/Protein Family
*LAP3*	4p15.32	5.00	Leucine aminopeptidase 3	M
*ERAP2*	5q15	4.40	Endoplasmic reticulum aminopeptidase 2	M
*METAP2*	12q22	4.22	Methionyl aminopeptidase 2	M
*TPP2*	13q33.1	4.13	Tripeptidyl peptidase 2	S
*DPP7*	9q34.3	4.04	Dipeptidyl peptidase 7	S
*ERAP1*	5q15	4.03	Endoplasmic reticulum aminopeptidase 1	M
*LTA4H*	12q23.1	3.44	Leukotriene A4 hydrolase	M
*LNPEP*	5q15	2.93	Leucyl and cystinyl aminopeptidase	M
*XPNPEP1*	10q25.1	2.66	X-prolyl aminopeptidase 1	M
*METAP1*	4q23	2.46	Methionyl aminopeptidase 1	M
*DPP3*	11q13.2	2.28	Dipeptidyl peptidase 3	M
*DPP8*	15q22.31	2.21	Dipeptidyl peptidase 8	S
*NPEPPS*	17q21.32	2.12	Aminopeptidase puromycin sensitive	M
*BLMH*	17q11.2	1.83	Bleomycin hydrolase	C
*JMJD7*	15q15.1	1.78	Jumonji domain containing 7	*
*RNPEP*	1q32.1	1.76	Arginyl aminopeptidase	M
*PGPEP1*	19p13.11	1.72	Pyroglutamyl-peptidase I	C
*TPP1*	11p15.4	1.59	Tripeptidyl peptidase 1	S
*PEPD*	19q13.11	1.41	Peptidase D	M
*DNPEP*	2q35	1.16	Aspartyl aminopeptidase	M
*DPP9*	19p13.3	1.16	Dipeptidyl peptidase 9	S
*METAP1D*	2q31.1	0.38	Methionyl aminopeptidase type 1D, mitochondrial	M
*NPEPL1*	20q13.32	0.37	Aminopeptidase like 1	M
*CTSH*	15q25.1	0.36	Cathepsin H	C
*XPNPEP3*	22q13.2	−0.04	X-prolyl aminopeptidase 3	M
*RNPEPL1*	2q37.3	−0.1	Arginyl aminopeptidase like 1	M
*AOPEP*	9q22.32	−0.38	Aminopeptidase O (putative)	M
*KDM8*	16p12.1	−1.3	Lysine demethylase 8	*
*ANPEP*	15q26.1	−2.13	Alanyl aminopeptidase, membrane	M
*ENPEP*	4q25	−2.15	Glutamyl Aminopeptidase	M
*MMP14*	14q11.2	−2.83	Matrix Metallopeptidase 14	M
*DPP4*	2q24.2	−2.84	Dipeptidyl Peptidase 4	S
*NAALADL1*	11q13.1	−3.39	N-Acetylated Alpha-Linked Acidic Dipeptidase Like 1	M
*AMZ1*	7p22.3	−3.6	Archaelysin Family Metallopeptidase 1	M
*LVRN*	5q23.1	−3.64	Laeverin	M
*CTSV*	9q22.33	−4.39	Cathepsin V	C
*XPNPEP2*	Xq26.1	−6.15	X-Prolyl Aminopeptidase 2	M
*F11*	4q35.2	−6.22	Coagulation Factor XI	S
*TRHDE*	12q21.1	−6.57	Thyrotropin Releasing Hormone Degrading Enzyme	M

^a^ Source of gene symbols and names: the HUGO Gene Nomenclature Committee (HGNC); RPKM: reads per kilobase of transcript per million mapped reads; C: cysteine peptidase; M: metallopeptidase; S: serine peptidase; *: Jumonji oxygenase family.

## Data Availability

The data presented in this study are available on request from the corresponding author. The data are not publicly available due to privacy and ethical limitations.

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
