# Peer review of "Aminopeptidase Expression in Multiple Myeloma Associates with Disease Progression and Sensitivity to Melflufen"

_cancers, 2021, doi:10.3390/cancers13071527_

Round 1
Reviewer 1 Report
The presented manuscript describes an increased expression of several aminopeptidases in relapsed/refractory myeloma, which is tested and validated on two quite large cohorts of MM patients. Interestingly, the increased expression is associated with increased CNV in aminopeptidase genes, suggesting potential novel mechanism of resistance development. Moreover, the increased expression of few of the tested aminopeptidases has an impact on the patients survival. While the expression of the aminopeptidases overall correlates with protein level of these genes, it is reather difficult to state if it correlates also with their high activity. Based on these findings, the authors used peptide conjugate drug melfufen to show that it induces stronger cytotoxicity in RRMM samples, likely due to increased expression of aminopeptidases. The results show that multiple aminopeptidases may be involved in this process.
The manuscript is well and clearly written, I have only one question:
In the figure 2, the authors show data related to aminopeptidase expression that is different between NDMM and RRMM at the level of adjusted p value <0.1. Why the authors chose this value, as commonly used adjusted p value to assess statistical significance is <0.05? Would the aminopeptidases be still differently expressed between the cohorts if the significance was determined at the level of adjusted p <0.05?
Author Response
Point 1.1: In the figure 2, the authors show data related to aminopeptidase expression that is different between NDMM and RRMM at the level of adjusted p value <0.1. Why the authors chose this value, as commonly used adjusted p value to assess statistical significance is <0.05?
Response 1.1: We thank the reviewer for the insightful question. The adjusted p-values were obtained after correction of the p-values for multiple testing using the Benjamini-Hochberg method through the DESeq2 tool. We used a significance lelvel of p<0.1 for the adjusted p-value (false discovery rate of 10%) to filter the differentially regulated gene by referring to the articles listed below:
1. Mallarino R, Henegar C, Mirasierra M, Manceau M, Schradin C, Valejo M, et al. Developmental mechanisms of stripe patterns in rodents. Nature. 2016;539(7630):518-23.
2. Tanaka S, Ise W, Inoue T, Ito A, Ono C, Shima Y, et al. Tet2 and Tet3 in B cells are required to repress CD86 and prevent autoimmunity. Nat Immunol. 2020;21(8):950-61.
3. Boomgarden AC, Sagewalker GD, Shah AC, Haider SD, Patel P, Wheeler HE, et al. Chronic circadian misalignment results in reduced longevity and large-scale changes in gene expression in Drosphila. BMC Genomics. 2019;20(1):14.
Point 1.2: Would the aminopeptidases be still differently expressed between the cohorts if the significance was determined at the level of adjusted p <0.05?
Response 1.2: To answer the reviewer’s second question, when the significance was determined at the level of adjusted p<0.05, three of the five aminopeptidases (DPP3, XPNPEP1, and ERAP2) previously identified at adjusted p<0.1 were still found to be differentially expressed (Figure 2A). The adjusted p-values of CTSV and RNPEP are 0.051 and 0.072, respectively, and therefore not regarded as differentially expressed when using an adjusted p<0.05.
Reviewer 2 Report
The present article by Drs. Miettinen et al. investigates the impact of aminopeptidase expression in multiple myeloma cells. They show similar expression patters in both the FIMM and the CoMMpass dataset; and that gene expression correlates with aminopeptidase protein levels. Moreover, authors provide evidence that elevation of aminopeptidase levels XPNPEP1, RNPEP, DPP3 and BLMH is associated with reduced survival. Conversely, data provide evidence that inhibition of aminopeptidases reduces cell viability of tumor cells. Finally, they aim to show that melflufen is dependent on aminopeptidase activity, LAP3, LTA4H, RNPEP and DPP7 in particular.
While investigations on the role of aminopeptidases and the peptide- conjugated drug melflufen in multiple myeloma are of interest, this study lacks a clear line. Many of the results are not conclusive. For example:
- Figure 1: based on hierarchical clustering the authors propose 4 groups among patients with ND and RRMM dependent on the expression levels of aminopeptidases. Which is the clinical relevance of these 4 groups, i.e. on survival?
- In vitro data on the anti-myeloma efficacy of melflufen in patients representative of the 4 groups should be depicted.
- The authors include data on patient cytogenetics. Did they observe a correlation between cytogenetics and expression levels of aminopeptidases in ND and RRMM patients?
- Figures 2 and 3: authors claim a statistically significant difference of aminopeptidases in ND versus RRMM samples, with XPNPEP1, RNPEP, DPP3, and CTSV being upregulated, and ERAP2 being downregulated. The correlation of XPNPEP1, RNPEP, DPP3 and survival needs to be shown separately in ND and RRMM samples. Which clinical relevance does ERAP2 downregulation in RRMM patients have?
- The authors indicate that survival curves in Figure 3B-E were generated in the CoMMpass dataset based on four aminopeptidase genes depicted from the FIMM dataset. Why did the authors include BLMH (Figure 3E), although it was not among the genes upregulated in Figure 2?
- Figure 4: The authors propose 3 patient groups based on melflufen- sensitivity of patient cells. Correlative data of the anti-myeloma activity and the presence of specific aminopeptidases in these 3 patient groups need to be included.
- Although the authors claim that myeloma cells are more sensitive towards melflufen than to melphalan (line 402 and 403), Figure 4B shows a highly significant correlation of melflufen and melphalan sensitivity in myeloma cells. It would be important to show correlative expression levels of aminopeptidases in these patients.
- The number of samples in Figure 4C is too small to derive final conclusions.
- Figure 5: Treatment with Melflufen > 3mcM for 15 minutes only is able to reduce cell viability to < 20%? Pharmacodynamics on myeloma cells need to be performed to support this finding. In addition, pharmacodynamics also need to be performed on healthy mononuclear cells 15 min up to 48hours.
- Rather than experiments presented in Fig. 5, genetic approaches need to be utilized to support the dependency of melflufen activity on distinct aminopeptidases.
- Figure 6: authors indicate that LAP3, LTA4H and RNPEP hydrolyze melflufen to melphalan. Previously, melflufen has been reported to be hydrolyzed by aminopeptidase N (ANPEP). How do the authors explain that ANPEP does not come up in their analyses?
Author Response
Response: Please see the attachment

Reviewer 3 Report
Miettinen et al. reported the association between aminopeptidase expression in CD138+ cells and disease progression in patients with multiple myeloma (MM). The findings are very interesting, although the following points should be addressed.
Comments:
- The authors successfully classified 4 groups according to the hierarchical clustering of aminopeptidase gene expression shown in Figure 1B. Is there any correlation between patients in group I and their clinical characteristics, i.e., serum M-protein levels, ISS stage, etc.?
- Five differentially expressed aminopeptidase genes in newly diagnosed and relapsed/refractory MM patients are shown in Figure 2B. Is there any correlation between the expression of each gene and patient characteristics? The sequential changes in the expression of those genes in the same patients should be described. P values for each gene should also be noted in the figure. Further, it would be preferable to show the expression levels of those 5 genes in plasma cells obtained from healthy volunteers or monoclonal gammopathy of undetermined significance patients.
- The number of patients in each group should be specified in Figure 3B–E.
- It is difficult to follow the text narrative in section 3.5. The authors should first explain how melflufen activity is dependent on aminopeptidase activity and then show the correlation between that activity and aminopeptidase gene expression in more detail.
Author Response
Point 1: The authors successfully classified 4 groups according to the hierarchical clustering of aminopeptidase gene expression shown in Figure 1B. Is there any correlation between patients in group I and their clinical characteristics, i.e., serum M-protein levels, ISS stage, etc.?
Response 1: In Figure 1B we have done hierarchical clustering of the aminopeptidase genes based on their gene expression levels and divided them into four groups. The patient samples haven’t been clustered in Figure 1B, but only divided to NDMM (on the left) and to RRMM (on the right). To try and answer the question presented to us, we performed correlation analysis to Figure 1B group 1 aminopeptidase genes in all 122 MM patient samples in the FIMM dataset using the following patient clinical characteristics: age, sex, serum M-protein levels, ISS and cytogenetics. We observed no correlation between the group 1 aminopeptidase genes and patient sex, serum M-protein levels or ISS. We did identify correlations with cytogenetics (Online Supplementary Figure S4A) and patient age (Online Supplementary Figure S4B). Online Supplementary Figure S4 has been added as an attachment (Please see the attachment). Patients having Del 13q have lower TPP2 (adjusted P=0.0240) gene expression compared to patients without Del 13q. Patients having t(4;14) have lower DPP7 (adjusted P<0.0001), ERAP1 (adjusted P=0.0005), LTA4H (adjusted P<0.0001) and LNPEP (adjusted P=0.0006) gene expression compared to patients without t(4;14). Patients with t(11;14) have lower expression of DPP7 (adjusted P=0.0780) and higher expression of LNPEP (adjusted P=0.0780) compared to patients without t(11;14). Patients having 1q gain have higher LAP3 (adjusted P=0.0003) gene expression compared to patients without 1q gain. Regarding patient age, patients <65 years of age have higher TPP2 (adjusted P=0.0170) and lower DPP7 (adjusted P=0.1) gene expression levels compared to patients ≥65 years of age. These results indicate that there are correlations between the group 1 aminopeptidase genes and certain patient clinical characteristics.
Point 2.1: Five differentially expressed aminopeptidase genes in newly diagnosed and relapsed/refractory MM patients are shown in Figure 2B. Is there any correlation between the expression of each gene and patient characteristics?
Response 2.1: We performed correlation analysis to the five aminopeptidase genes shown in Figure 2B in 122 MM patient samples in the FIMM dataset using the following patient clinical characteristics: age, sex, serum M-protein levels, ISS and cytogenetics. We observed no correlation between the five aminopeptidase genes shown in Figure 2B and patient sex, age and serum M-protein levels. We did identify correlations with patient cytogenetics (Online Supplementary Figure S7A) and ISS (Online Supplementary Figure S7B). The Online Supplementary Figure S7 has been added as an attachment (Please see the attachment). Patients with Del13q have higher RNPEP expression (adjusted P=0.075) compared to patients without Del13q. Patients with t(4;14) have higher expression of RNPEP (adjusted P=0.012) and CTSV (adjusted P<0.0001) compared to patients without t(4;14). Patients with t(11;14) have higher expression of ERAP2 (adjusted P=0.0720), lower DPP3 (adjusted P=0.0096) and CTSV (adjusted P=0.0096) expression compared to patients without t(11;14). Patients with 1q gain have higher expression of XPNPEP1 (adjusted P=0.0220) and RNPEP (adjusted P=0.0002) compared to patients without 1q gain. Regarding differences based on ISS stage, ISS 3 (adjusted P=0.0559) have higher expression of RNPEP compared to ISS 1. Patients with ISS 3 (adjusted P=0.0479) have also higher expression of RNPEP compared to ISS 2. ISS 3 (adjusted P=0.0807) have higher expression of XPNPEP1 compared to ISS 2. ISS 3 (adjusted P=0.0559) have higher expression of DPP3 in compared to ISS 2. These results indicate that there are some correlations between the five aminopeptidase genes associated with differential expression in NDMM and RRMM and certain patient clinical characteristics.
Point 2.2: The sequential changes in the expression of those genes shown in Figure 2B in the same patients should be described.
Response 2.2: For the samples shown in Figure 2B this can’t be described as there are only 2 patients with sequential samples including both a NDMM and RRMM sample in the FIMM dataset. The CoMMpass dataset which we have used for data validation purposes in the manuscript has sequential samples from 39 patients (NDMM = 39, RRMM = 45). We have used these CoMMpass datasetpaired samples already for the comparison of the five aminopeptidase genes identified in the FIMM dataset to be differentially expressed between NDMM and RRMM (results shown in Online Supplementary Figure 6A). We have now described the sequential changes in the gene expression of the five aminopeptidase genes shown in Figure 2B using the CoMMpass dataset paired samples (Online Supplementary Figure S6B). The Online Supplementary Figure S6B has been added as an attachment (Please see the attachment). The results support the aminopeptidase gene expression differences identified between NDMM and RRMM samples using the sample cohorts from FIMM (Figure 2B) and CoMMpass (Online Supplementary Figure S6A) datasets. The results also show that there are some patients in which the aminopeptidase gene expression doesn’t follow the trend indicated in Figure 2B and Online Supplementary Figure S6A as can be expected.
Point 2.3: P values for each gene should also be noted in the figure 2B.
Response 2.3: Both the non-adjusted p-values and adjusted p-values have now been added to the revised manuscript Figure 2B.
Point 2.4: Further, it would be preferable to show the expression levels of those 5 genes in plasma cells obtained from healthy volunteers or monoclonal gammopathy of undetermined significance (MGUS) patients.
Response 2.4: Although we agree that this would be a useful analysis, we only have gene expression data available from CD138+ cells isolated from two healthy donors. However, we found that data from these two healthy donor samples was not sufficient to make any conclusions on gene expression differences between healthy and myeloma plasma cells. In addition, we do not have access to MGUS patient gene expression data, and are therefore unable to perform the suggested analysis to determine if aminopeptidase expression differs between MGUS and MM patients. Thus, we are not able to show aminopeptidase gene expression levels from plasma cells collected from healthy donors or MGUS patients for the five genes shown in Figure 2B.
Point 3: The number of patients in each group should be specified in Figure 3B–E.
Response 3: The number of patient samples in each group is now indicated in the revised manuscript Figure 3. The numbers of high and low gene expression patients are the same in each of the figures 3B-E.
Point 4: It is difficult to follow the text narrative in section 3.5. The authors should first explain how melflufen activity is dependent on aminopeptidase activity and then show the correlation between that activity and aminopeptidase gene expression in more detail.
Response 4: We have modified section 3.5 in the revised manuscript to make it easier for the reader to follow the text narrative.

Round 2
Reviewer 2 Report
The authors addressed some of my own and reviewer 3’s concerns by including additional analyses. However, authors’ responses to my comments regarding experiments in Figures 4 to 6 remain to be insufficient.
Author Response
Please see the attachmen

Reviewer 3 Report
The authors responded well to the comments in this revised version. These findings are very interesting and informative for readers.
Author Response
We would like to thank Reviewer 3 for their constructive comments which improved the manuscript, and for accepting our responses to those comments.